# The Interaction between the Host Genome, Epigenome, and the Gut–Skin Axis Microbiome in Atopic Dermatitis

**DOI:** 10.3390/ijms241814322

**Published:** 2023-09-20

**Authors:** Rodrigo Pessôa, Patricia Bianca Clissa, Sabri Saeed Sanabani

**Affiliations:** 1Postgraduate Program in Translational Medicine, Department of Medicine, Federal University of Sao Paulo (UNIFESP), Sao Paulo 04039-002, Brazil; rodrigo_pessoa1@hotmail.com; 2Immunopathology Laboratory, Butantan Institute, Sao Paulo 05503-900, Brazil; patricia.clissa@butantan.gov.br; 3Laboratory of Medical Investigation LIM-56, Division of Dermatology, Medical School, University of Sao Paulo, Sao Paulo 05508-220, Brazil; 4Laboratory of Medical Investigation Unit 03, Clinics Hospital, Faculty of Medicine, University of Sao Paulo, Sao Paulo 05403-000, Brazil; 5Laboratory of Dermatology and Immunodeficiency LIM56/03, Instituto de Medicina Tropical de Sao Paulo, Faculdade de Medicina, University of Sao Paulo, Av. Dr. Eneas de Carvalho Aguiar, 470 3º Andar, Sao Paulo 05403-000, Brazil

**Keywords:** microbial dysbiosis, intestinal permeability, immuno-regulation, short-chain fatty acid, virome, single nucleotide polymorphism, genetics, epigenetics

## Abstract

Atopic dermatitis (AD) is a chronic inflammatory skin disease that occurs in genetically predisposed individuals. It involves complex interactions among the host immune system, environmental factors (such as skin barrier dysfunction), and microbial dysbiosis. Genome-wide association studies (GWAS) have identified AD risk alleles; however, the associated environmental factors remain largely unknown. Recent evidence suggests that altered microbiota composition (dysbiosis) in the skin and gut may contribute to the pathogenesis of AD. Examples of environmental factors that contribute to skin barrier dysfunction and microbial dysbiosis in AD include allergens, irritants, pollution, and microbial exposure. Studies have reported alterations in the gut microbiome structure in patients with AD compared to control subjects, characterized by increased abundance of *Clostridium difficile* and decreased abundance of short-chain fatty acid (SCFA)-producing bacteria such as *Bifidobacterium*. SCFAs play a critical role in maintaining host health, and reduced SCFA production may lead to intestinal inflammation in AD patients. The specific mechanisms through which dysbiotic bacteria and their metabolites interact with the host genome and epigenome to cause autoimmunity in AD are still unknown. By understanding the combination of environmental factors, such as gut microbiota, the genetic and epigenetic determinants that are associated with the development of autoantibodies may help unravel the pathophysiology of the disease. This review aims to elucidate the interactions between the immune system, susceptibility genes, epigenetic factors, and the gut microbiome in the development of AD.

## 1. Introduction

Atopic dermatitis (AD), also known as atopic eczema, is a common inflammatory skin disease affecting 7–10% of adults and up to 25% of young children [1,2]. AD results from a complex interaction between the host immune system and various environmental factors in genetically susceptible individuals [3,4,5]. The prevalence of AD in different genders during childhood and adolescence remains inconclusive. According to some authors, the natural history of AD suggests that males predominate in childhood and females after puberty [6]. Studies in Europe have suggested an increase in the incidence and prevalence of AD in the 21st century compared to the 20th century [7,8], possibly due to various factors such as genetic and epigenetic factors, impaired skin barrier integrity, autoimmunity, viral infections, gut microbiome composition, dietary habits, and lifestyle factors [7,9,10,11,12,13,14,15,16]. Recent research has shown functional links between genetic variants in the IL-17 promoter, the gut microbiome, and AD [17]. The A allele of the IL-17 mutation rs2275913 has been associated with higher expression of IL-17 and dysbiosis, leading to intestinal and systemic inflammation and early onset of AD. The hygiene hypothesis has been proposed to explain the significant increase in AD in developing countries, where lack of exposure to infectious agents may affect immune maturation [18]. Although significant progress has been made in understanding the pathophysiology of AD, further research is needed to translate this knowledge into clinical decisions. In this article, we aim to explore the interaction between host genetic and epigenomic factors and the gut microbiome in AD.

## 2. Genetic Predisposition to AD

Esparza-Gordillo et al. published the first AD genome-wide association study (GWAS) in May 2009. The study included 939 cases, 975 German controls, and 275 nuclear families, each having two affected siblings [19]. This study confirmed that the filaggrin locus (FLG) is a risk factor for AD and found a novel susceptibility region on chromosome 11q13.5, 38 kb upstream of C11orf30. In 2011, Sun et al. published the results of a GWAS in the Chinese Han population [20]. In this study, the FLG region was validated in the Chinese population, and two novel loci were discovered at 5q22.1 and 20q13.33. The validation of these two loci was performed in 1806 cases and 3256 controls from Germany. A meta-analysis of GWAS in Europe by Paternoster et al. identified three additional novel risk loci for AD (11q31.1, 19p13.2, 5q31) [21]. In 2015, Paternoster et al. conducted a multi-ancestry GWAS of the AD cohort with the best statistical power to date, including 21,000 patients and 95,000 controls, and validated the results in 32,059 cases and 228,648 controls [22]. This analysis confirmed previously reported AD risk loci and identified 11 additional new AD risk loci. The newly discovered loci include candidate genes for CD207 (langerin), PPP2R3C, IL-7R, STAT3, and ZBTB10, known to be involved in T-cell control and innate host defense. In another study by Esparza-Gordillo et al., the association of IL6R rs2228145 (C) genotype with higher plasma levels of soluble IL-6R in AD and sustained AD status was confirmed using two independent population-based cohorts [23]. Heritability studies demonstrate the importance of both general genetic factors related to atopy and disease-specific AD genes. According to studies in twins, the heritability of AD ranges from 71% to 84% [24]. However, these genetic factors are not 100% predictive of AD, suggesting a complex interaction between genetic determinants and environmental factors such as the gut microbiota [13,17].

## 3. Gut Microbiome, Immunity, and AD

The gut microbiota is a diverse consortium of microorganisms that reside in the gastrointestinal tract, comprising bacteria, viruses, fungi, protozoa, and archaea [25]. It encompasses approximately 500–1000 different species, with the dominant phyla in healthy individuals being *Bacteroidetes*, *Firmicutes*, Actinobacteria, and *Proteobacteria* [26,27,28,29].The microbiome exerts various beneficial effects, including influencing energy balance, metabolism, gut epithelial cell health, immunologic activity, neurodevelopment, vitamin synthesis, and immune system maturation [26,30]. Additionally, byproducts of the gut microbiota can modulate host physiology and metabolism, aiding digestion and energy production from indigestible substrates. One example is the extraction of short-chain fatty acids (SCFAs) from indigestible fiber [26,27]. SCFAs fuel the intestinal mucosa, regulating immune responses and maintaining intestinal homeostasis to prevent inflammation and carcinogenesis [31,32].The composition of the gut microbiota is influenced by various factors such as changes in diet, increased stress, lifestyle behaviors, sex hormones, genetic makeup, and prolonged antibiotic use [33]. Alterations in the gut microbial composition have been implicated in the development of several diseases, including gastrointestinal disorders, cardiovascular conditions, atopic dermatitis (AD), diabetes, and obesity [13,34,35,36,37]. Remarkably, numerous studies have compared the microbiomes of individuals with AD or those genetically predisposed to the condition to that of healthy controls, revealing differences and suggesting the involvement of dysbiotic gut microbiota in the disease’s development [38,39,40].

### 3.1. Mode of Delivery and the Infant’s Gut Microbiota: Impact on the Risk of AD

Several studies have shown that the mode of delivery affects the composition of the infant’s gut microbiota. Vaginal delivery (VD) contributes to the normal colonization of the infant’s gut by exposing them to maternal vaginal microbiota, which includes *Lactobacillus*, *Prevotella*, *Bacteroides*, *Escherichia*, *Shigella*, and *Bifidobacterium* [41,42]. In contrast, delivery by cesarean section (CD) has been associated with delayed acquisition of vaginal microbiota, such as Bacteroides species. This delay has been linked to lower levels of Th-1-associated chemokines CXCL10 and CXCL11 in infants’ blood, suggesting the importance of this microbiota in promoting cytokine synthesis necessary for neonatal immunity [43,44]. Epidemiological evidence indicates that infants born via CD are more likely to develop AD compared to those born via VD [45,46,47]. Thus, the structure of the gut microbiome in early life appears critical for long-term health. It is crucial to gain a more precise understanding of neonatal gut ecology. Recent findings have noted an association between CD and reduced levels of microbial metabolites like riboflavin and folate. This association suggests that impaired folate biosynthesis may affect the immune function of natural killer cells against viral infections, potentially triggering AD [48]. Furthermore, antibiotic use during pregnancy and the early postnatal period elevates the risk of AD in infants [49]. These studies collectively highlight the critical role of a healthy microbiome in maintaining a strong immune system during early life.

### 3.2. Environmental Factors, the Gut Microbiome, and AD

It is generally believed that hereditary variables, such as a history of eczema in parents, influence the prevalence of eczema in their offspring [50,51]. Additionally, environmental factors have been shown to increase disease prevalence by affecting genetic predisposition [52,53]. Common allergens known to trigger AD symptoms include pollen, dust mites, and animal dander [54]. Extensive research has been conducted in recent years to identify environmental elements that increase the risk of AD and to find interventions to reduce or prevent the disease. Epidemiological data suggest a higher prevalence of AD in wealthy industrialized countries compared to developing countries [55,56,57,58]. This is believed to be attributed to stringent hygiene practices in developed countries, which limit exposure to beneficial microbes, impacting the education of the host immune system [51,59]. The cause of allergic diseases, including AD, is largely associated with abnormal immunological activation and reactivity, particularly in relation to early microbial exposure. This is especially relevant given the high prevalence of AD in younger age groups. Additionally, certain genes involved in the immune system may influence prenatal and neonatal interaction with intestinal bacteria [60]. The development of an individual’s gut flora, which influences susceptibility to AD, primarily occurs during infancy and early childhood. The gut microbiome may play a critical role in AD development by regulating the maturation of the immune system through interactions with the host, especially during infancy and early childhood [61,62,63]. Studies have reported increased abundance of *Clostridium difficile*, *Escherichia coli*, and *Staphylococcus aureus* (*S. aureus*), as well as decreased abundance of *Bifidobacteria* and *Bacteroides*, in the gut microbiome of AD patients compared to healthy controls (Figure 1). For instance, Watanabe et al. found significantly lower levels of *Bifidobacteria* in AD patients compared to healthy individuals [38]. Moreover, the abundance of *bifidobacteria* varies according to disease severity, with severe AD patients having lower levels compared to those with moderate atopy. A recent study by Fieten et al. identified a microbial signature that distinguishes between AD children with and without food allergies. AD children with food allergies showed increased levels of *Bifidobacterium pseudocatenulatum* (*B. pseudocatenulatum*) and *Escherichia coli* in their feces, but decreased levels of *Bifidobacterium adolescentis* (*B. adolescentis*), *Bifidobacterium breve* (*B. breve*), *Faecalibacterium prausnitzii* (*F. prausnitzii*), and *Akkermansia muciniphila* [40]. Additionally, several cohort studies have provided evidence that disrupted gut flora may precede the development of AD. Infants with mild AD or healthy infants have been found to have a higher prevalence of butyrate-producing bacteria such as *Coprococcus eutactus* compared to infants with severe AD [64]. *Coprococcus* are bacteria that produce SCFAs, along with *Bifidobacterium*, *Blautia*, *Eubacterium*, and *Propionibacterium*, all of which have been detected in lower amounts in individuals with AD compared to those without AD [65]. SCFAs such as butyrate, propionate, and acetate are critical for maintaining host health by exerting anti-inflammatory effects through multiple mechanisms, including the maintenance of the mucus layer and epithelial integrity [66]. Therefore, decreased production of SCFAs may be responsible for the observed intestinal barrier breakdown, increased intestinal permeability, and inflammation in AD [67]. In the large intestine of healthy humans, acetate is produced by *Bifidobacteria* and certain *Firmicutes* and *Bacteroidetes* species [68], while propionate is produced by *Bacteroidetes*, *Negativicutes* (*Firmicutes* strain), and *Lachnospiraceae* [69]. Butyrate is produced by *Faecalibacterium prausnitzii, Eubacterium rectale*, *Eubacterium hallii* [70]. Reduced species diversity of the intestinal flora and delayed colonization with *Bacteroidetes* have also been associated with AD, particularly in infants at one month of age [63]. According to the concept of “microbial deprivation syndromes of affluence”, lower intensity and diversity of microbial stimulation during early childhood inhibit Th1 induction and Th2 suppression [71]. These results suggest a possible link between AD and a Th2-type immune response to skin allergens triggered by a dysbiotic gut microbiota, dysregulated gut inflammation, and a disrupted epithelial barrier. The gut–skin axis, which refers to the influence of gut flora on bacteria residing on the skin, plays a crucial role in AD. Although the exact mechanism remains unclear, researchers suspect that an imbalanced microbiota contributes to the inflammation and immunological responses observed in AD. Fecal microbiota transplantation (FMT) is the preferred approach for restoring the balance of the gut microbiota compared to probiotics. This is because FMT involves transferring the entire functional community of the gut microbiota from healthy donors to recipients [72]. FMT has shown effectiveness in treating various clinical conditions, including gastroenterological, metabolic, and autoimmune diseases [73]. Limited evidence from experimental studies in both humans and mice suggests that FMT can restore the gut microbiota and immune balance (Th1/Th2) while also suppressing allergic reactions associated with AD [74,75,76]. As a result, FMT holds promise as a potential new therapy for AD. Additionally, there is evidence that dietary interventions, such as probiotics and prebiotics, can modulate the gut microbiome and improve symptoms in individuals with AD [77,78]. In AD, there is a hypothesis of molecular mimicry occurring between environmental allergens, such as pollen or dust mites, and self-antigens in the skin [79,80]. For instance, the protein profilaggrin, vital for maintaining the integrity of the skin barrier, has been identified as a potential self-antigen that the immune system of individuals with AD may target [81]. Studies have shown that allergens, such as dust mites and pollen, contain molecules that are structurally similar to profilaggrin and other skin proteins [82,83]. When these allergens come into contact with the skin, they can trigger an immune response against both the allergen and the self-antigen, leading to inflammation and tissue damage [84].

### 3.3. Gut Virome, Mycobiome, and AD

The gut virome plays a crucial role in maintaining a healthy immune system by modulating the production of cytokines, which are important signaling molecules involved in inflammation and immune responses. In a recent study by Xiang et al. [13], the composition of the gut virome in 21 AD patients and 12 healthy controls was examined to better understand the effects of the gut microenvironment on AD. The results of this study did not reveal a clear pattern distinguishing the characteristics of the viral intestinal community between the two groups. Regarding diversity, the study confirmed that the alpha diversity of the AD patient group was significantly lower than that of the healthy control group, and the beta diversity showed significant differences between the groups. Gut mycobiomes in AD were investigated in a unique study by Mok and colleagues [85] in a group of 9- to 12-month-old infants who were divided into two groups based on their symptoms (cured or still experiencing symptoms). The authors employed metagenomic and metaproteomic approaches and concluded that mycobiome diversity was higher in the group still experiencing symptoms. Infants with gastrointestinal dysbiosis (GI) showed an increase in *Rhodotorula* sp. and a decrease in the *Ascomycota*/*Basidiomycota* ratio, whereas *Wickerhamomyces* and *Kodamaea species* increased significantly in the healthy group. Interestingly, microorganisms of the genera *Acremonium* and *Rhizopus* were more abundant in the healthy group. Additionally, the authors identified five fungi as biomarkers for each sample group and utilized a metaproteomic approach to determine that the diseased cohort had a greater amount of fungal proteins, with *Rhodotorula* sp. being the major producer. Moreover, this yeast fungus, which is widely distributed in the environment, produces two unique proteins, RAN-binding protein 1 and glycerol kinase, which are specific to infants with AD, suggesting their potential involvement in the development of this syndrome. However, further studies are needed to fully understand the role of the gut mycobiome in AD, including its interactions with the gut microbiome and the immune system.

## 4. Microbial Metabolites, Probiotics, and AD

Gut microorganisms produce a variety of chemicals or metabolites that can affect a range of host activities. Several studies to date have shown that only certain categories of metabolites have been identified and thoroughly investigated. Of greatest importance are those that exert an appreciable influence on immune system function. These include SCFAs, tryptophan metabolites, and amine derivatives, among which trimethylamine N-oxide (TMAO) stands out (Figure 2).

### 4.1. Microbial Metabolites and Their Potential Role in AD

SCFAs, which are metabolites produced by intestinal bacteria through the fermentation of dietary fiber, are the most extensively studied substances among gut bacterial metabolites. They are known to have anti-inflammatory properties and may help regulate the immune system. Studies have shown that butyrate can inhibit NF-κB activation in intestinal epithelial cells, immune cells, and other cell types of the gut [87]. By blocking NF-κB signaling, butyrate can decrease the expression of pro-inflammatory cytokines [88], and chemokines. Within 24 h, butyrate significantly decreased the levels of chemokines CCL2 and CCL5. It can also promote the expression of anti-inflammatory factors [89,90]. In contrast to previous studies, SCFAs were found to enhance Th1 and Th17 cell generation during ongoing immune responses while promoting cytotoxic activity and IL-17 production of CD8 T cells [91]. A possible explanation for these discrepancies might be attributed to the host’s condition with SCFAs displaying tolerogenic properties in steady states but stimulating a more effective immune response during acute immune responses to combat infections [92].

Protein breakdown occurs in the gastrointestinal tract and results in the formation of amino acids, including tryptophan. Much evidence suggests that intestinal bacteria such as *Clostridium*, *Bacteroides*, *Bifidobacterium*, and *Lactobacillus* metabolize tryptophan, primarily via indole pathways, and that this metabolism affects host health [93,94]. Although the compounds formed in the indole pathway serve several functions, such as enhancing GLP-1 production and promoting intestinal motility, their main importance in the field of dermatological diseases seems to be the activation of the aryl hydrocarbon receptor (AhR) [94,95]. The activation of this receptor in keratinocytes from patients with AD has been associated with the upregulation of filaggrin and loricrin, which are important proteins that build the skin barrier [96]. The importance of the AhR pathway in dermatology became apparent with the discovery of the innovative drug tapinarof, which acts as an AhR agonist. Tapinarof is currently being studied in several phase 3 clinical trials in both adults and children with AD [97,98]. The encouraging results observed with AhR activation suggest that tryptophan metabolites may have comparable therapeutic utility. They could potentially serve as valuable adjuncts in the treatment of skin diseases.

Trimethylamine (TMA) and trimethylamine N-oxide (TMAO) are biologically active compounds formed when gut microflora (*Firmicutes*; especially the *Clostridium cluster XIVa* and *Eubacterium*, *Actinobacteria,* and *Proteobacteria*) metabolize certain small molecules containing a quaternary amino group such as choline, L-carnitine, or phosphatidylcholine [99]. These substances are often found in foods such as eggs, liver, dairy products, and peanuts. By activating the NLRP3 inflammasome, TMAO plays a role in increasing the levels of proinflammatory cytokine; IL-1β, IL-6, and IL-18 and has been implicated in a variety of inflammatory diseases including psoriasis disorder [92,100]. Currently, there is a lack of research on TMAO levels in AD. Although TMAO has shown promise in studies of other inflammatory diseases, further investigation is needed to fully elucidate the involvement of this bacterial metabolite in pathogenesis AD and translate this knowledge into clinical applications.

### 4.2. Prebiotics and Probiotics in AD

In recent years, there has been an increase in the number of research papers on the effects of prebiotics and probiotics on symptoms and disease severity in patients with AD. Improving gut barrier function is one of the mechanisms by which probiotics alter the gut microbiome and the immune state of the host. These effects are responsible for attenuating allergic reactions and reducing AD severity [101]. The most popular probiotics used in randomized clinical trials for the treatment of AD are *lactobacilli* and *bifidobacteria*, both found in dairy products. These bacteria can modulate immune cells to restore TH1/TH2 immune balance, increase the production of the regulatory cytokine IL-10 [102], and increase the population of Tregs [103], all of which contribute to the treatment of AD. They also compete with pathogenic bacteria, including *S. aureus* which is associated with AD, for nutrients and mucin binding [104]. In contrast, when KM-H2 cells were exposed to D-tryptophan, a metabolite associated with *Bifidobacterium*, *Lactobacillus*, and *Lactococcus*, there was a reduction in the presence of Th2-associated CCL17 [105]. Administration of a single probiotic strain has been found to be effective and durable in preventing the disease [104,106,107,108]. It has been demonstrated that infants who received *L. rhamnosus* GG (LGG) during the first two years of life had a reduced risk of developing AD by 50% [107,109]. According to the results of a large-scale randomized clinical trial conducted in New Zealand, early exposure to *L. rhamnosus* (HN001) was associated with long-term protection against AD, especially during the first ten years of life [108]. These results were comparable to those of Kalliomaki et al. [107], who used a similar probiotic (*L. rhamnosus* GG) and found a significant reduction in relative risk of 49%, 43%, and 36% at ages 2, 4, and 7 years, respectively. A mixture of probiotics also appears to be useful in preventing AD. In a Korean study [110], a probiotic mixture of *B. bifidum* BGN4, *B. animalis* subsp. *lactis AD011*, and *L. acidophilus AD031* resulted in a significant reduction in AD incidence in the first year of life. In other studies conducted in Norway [111,112], maternal intake of a mixture of three probiotics, *LGG*, *L. acidophilus La-5*, and *B. lactis Bb-12*, was found to have a significant preventive effect against the development of AD over a period of up to six years, associated with a decrease in TH22. Overall, the systematic reviews and meta-analyses conclude that probiotics have a long-term preventive effect against AD. Therefore, despite conflicting data, the World Allergy Organization recommends the use of probiotics during pregnancy, lactation, and infancy for the prevention of allergies [113].

## 5. The Role of Genetic Predisposition in the Gut Microbiome Composition of AD Patients

Although genetic studies have identified AD-related genes (e.g., IL4, IL4R, IL13, CMA1, SPINK5, FLG, IL-6) [114,115], the predicted genetic contribution does not provide a 100% predictive value, indicating the likely role of other environmental variables, including the gut microbiota. Numerous studies have explored the effects of host genetics on the gut microbiome, particularly in the gut. Studies comparing the gut microbiomes of genetically related individuals (e.g., twins) with those of unrelated individuals have shown strong correlations between genetic relatedness and gut microbiome composition [116,117]. However, these correlations have not been consistently found in studies involving twins or larger population cohorts [118]. Some studies have identified significant effects of genetic host factors on the abundance of specific gut taxa [118,119], while other population cohort studies have not found such effects [120]. Further research is needed to fully understand the complex interactions between genetics, the gut microbiome, and the development of AD. This knowledge could ultimately lead to the development of novel therapeutic approaches for this complex condition.

## 6. The Role of the Gut Microbiome in Gene Expression and Epigenetic Regulation of AD

As mentioned above, the occurrence and progression of AD are closely linked to the interaction of genetic predisposition and environmental variables. One of the most important aspects of this interaction is epigenetic regulation, which involves histone modifications, DNA methylation, and noncoding RNA binding [121,122,123,124]. Dysbiosis of the gut microbiome and its metabolites have been implicated in triggering these epigenetic interactions. Specifically, these metabolites have been shown to influence miRNA expression variations in several genes associated with AD, including the chemokine receptor CCR7 and HAS3 [125,126,127].

### 6.1. Non-Coding RNA Binding

Non-coding RNA (ncRNA), including miRNA, is a class of RNA transcripts that does not code for proteins and has been associated with the risk of AD [128]. Many miRNAs have been implicated in the etiology of AD, suggesting an epigenetic role for these molecules. Furthermore, the link between the gut microbiota and immune function outside the gut goes beyond the extraintestinal immune response. It has been reported that the gut microbiota can also influence the expression of host extraintestinal miRNAs, termed the “microbiome–miRNA axis” [129] as shown in Figure 3. In the context of AD, dysregulated miRNA expression patterns were observed in affected skin tissues, blood samples, and even in the lesional and non-lesional skin of AD patients. These dysregulated miRNAs were found to target genes involved in immune regulation, inflammatory signaling pathways, and skin barrier maintenance. For instance, overexpression of miR-155 and other miRNAs has been observed in immune cells from AD patients [130]. It has a pathogenic effect by modulating cytotoxic T lymphocyte antigen-4 (CTLA-4), a negative regulator of T cell activation [131]. Another miRNA involved in the development of AD is miR-146a [132]. The keratinocytes of AD patients exhibit elevated levels of miR-146a [133]. By inhibiting the expression of NF-κB upstream elements such as CARD10 and IRAK1, this miRNA suppresses the production of NF-κB-dependent genes in human primary keratinocytes, thereby reducing IFN-gamma and NF-κB-activated chronic skin inflammation [133,134]. miR-124a has been shown to target NF-κB, which is required for the formation of pro-inflammatory chemokines in activated keratinocytes. A recent study by Yang et al. [135] showed lower expression of miR-124a in lesional tissues of AD patients, suggesting that miR-124a may reduce chronic skin inflammation in AD by inhibiting innate immune responses in keratinocytes. According to recent studies, there is a strong correlation between certain bacterial families, such as *Enterobacteriaceae*, miR-194-5p, and let-7c-5p in the gut [136]. These findings suggest that the intestinal microbiota may alter the profile of fecal miRNAs, which can mediate host-microbiota interactions and regulate intestinal health [136]. Currently, there are no data to investigate the possible relationship between gut dysbiosis, fecal miRNA composition, and SCFA levels in response to AD intestinal inflammation.

### 6.2. DNA Methylation

Bacteria, including those in the gut, have the potential to alter the host epigenome by producing epigenetically active chemicals such as SCFAs, which are required for the regulation of DNA methylation, or by activating cell signaling pathways that influence epigenetic gene expression [137]. Several species of commensal skin bacteria such as *S. epidermidis*, *P. acnes*, and the pathogenic *S. aureus* are capable of fermenting a common carbon source such as glucose to SCFA [138]. Epigenetic mechanisms, particularly methylation, are critical for immune control and are influenced by a range of environmental stimuli that induce molecular changes in genes. Methylation involves the addition of a methyl group to cytosine (C5 position; 5-methylcytosine, 5mC), primarily occurring in the context of CpG dinucleotides [139,140]. In AD research, DNA methyltransferase-1 (DNMT-1) has been the primary enzyme of interest. Nakamura et al. [141] performed an indirect assessment of methylation status in patients with AD by quantifying DNMT-1 mRNA expression in peripheral blood mononuclear cells using RT-PCR. They observed lower DNMT-1 mRNA expression in patients with AD compared to healthy controls, although the difference was not statistically significant. However, the study also considered that IgE levels in the cohort of patients with AD, and individuals with high IgE levels exhibited significantly lower DNMT-1 mRNA levels.

## 7. Summary

In the past few years, significant advancements have been made in understanding the role of the host genome, epigenome, and the gut–skin axis microbiome in AD. These areas of research have provided valuable insights into the pathogenesis and potential therapeutic approaches for this complex skin disorder. Studies have focused on identifying genetic variations associated with AD susceptibility and severity. The GWAS have identified several genetic loci that are linked to AD, including FLG gene mutations, which are the most well-established risk factor for AD [142]. Recent studies have further explored the genetic architecture of AD, uncovering additional susceptibility loci and gene–environment interactions [143]. These findings have improved our understanding of the underlying genetic mechanisms involved in AD development. Epigenetic modifications, including DNA methylation, histone modifications, and non-coding RNA expression, have been investigated to understand their role in AD. Epigenetic changes can regulate gene expression and contribute to disease susceptibility and progression. Recent studies have identified specific DNA methylation patterns associated with AD, highlighting the importance of epigenetic regulation in disease pathogenesis [144,145]. These different CpG methylation patterns showed partial correlation with genes that are differentially regulated and play a role in epidermal differentiation and innate immune response. Differential methylation has been shown to be an important factor highlighting the involvement of DNA methyltransferases responsible for attaching methyl groups to genes. In a recent study by Yoshida and coworkers [146] examining both human and mouse models of AD, the researchers observed decreased levels of DNMT1in individuals with AD. Interestingly, AD patients with lower DNMT1 expression exhibited more severe itch than patients with relatively higher DNMT1 expression. In addition, epigenetic alterations have been explored as potential therapeutic targets for AD [147]. The gut–skin axis represents the bidirectional communication between the gut microbiome and the skin [148]. Recent research has highlighted the role of gut dysbiosis in AD pathogenesis [149]. Dysregulation of the gut microbiome composition and function has been associated with AD development and severity [150]. Available data have demonstrated alterations in microbial diversity among patients with AD, characterized by an elevated prevalence of *Clostridium difficile*, *Escherichia coli*, and *S. aureus* in comparison to healthy individuals. Conversely, the gut microbiome of AD patients exhibits a reduced colonization of *Bifidobacteria*, *Bacteroidetes*, and *Bacteroides* when compared to healthy controls [151,152,153,154]. Modulation of the gut microbiota through dietary interventions, prebiotics, probiotics, and fecal microbiota transplantation has shown promise as a therapeutic approach for AD.

## 8. Conclusions

This review provides an overview of recent studies investigating the impact of the gut microbiota and epigenetics on the development and progression of AD. While there is evidence of interactions between the host genome and AD, as well as between the gut microbiome and AD, it remains uncertain whether dysbiotic microbiomes are genetically determined. Further investigation is needed to elucidate the molecular mechanisms underlying the protective effects of certain bacterial species and metabolites against AD. The implementation of different molecular approaches holds great promise in expediting the discovery of novel causes and potential treatments for AD.

## Figures and Tables

**Figure 1 ijms-24-14322-f001:**
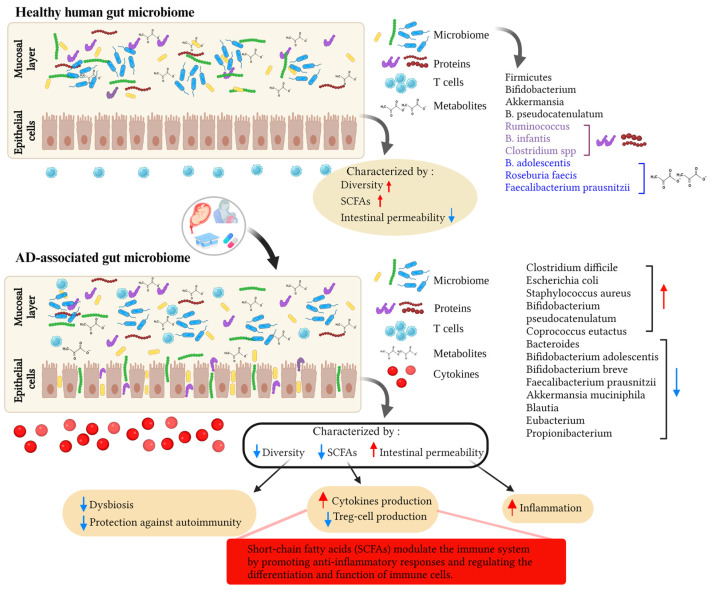
This figure illustrates the effects of environmental factors on the gut microbiota and their potential role in the development of atopic dermatitis (AD). Various environmental factors, including diet, hygiene, allergen exposure, and antibiotic use, can influence the composition and diversity of the gut microbiota. Alterations in the composition of the gut microbiota can disrupt gut barrier function and allow translocation of microbial metabolites and activation of the immune system. These processes contribute to the pathogenesis of AD and highlight the complex interplay of environmental factors, gut microbiota, and AD development. The red arrow pointing up means an increase, the blue arrow pointing down means a decrease. Created with BioRender.com.

**Figure 2 ijms-24-14322-f002:**
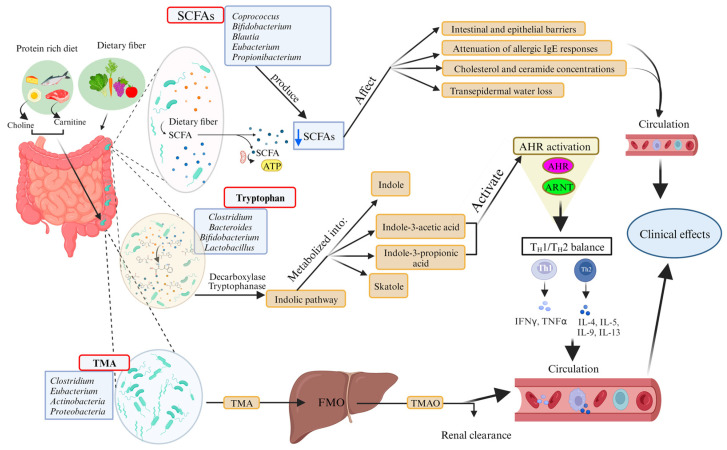
The gut microbiome functions by producing active metabolites that can affect AD. Metabolites/pathways of the gut microbiome have been described. First, dietary fiber is converted by the gut microbiome into short-chain fatty acids (SCFAs, red box), which include acetic acid, propionic acid, and butyric acid. The microbiota in AD produces fewer SCFAs, and this low SCFA content has an effect not only on improving the intestinal and epithelial barrier, but also on regulating immune cell migration to sites of inflammation in AD lesions, as well as modulating allergic IgE responses, cholesterol and ceramide concentrations, and controlling trans-epidermal water loss to maintain the epidermal barrier in AD. The microbial metabolite D-tryptophan (red box) can also restore TH1/TH2 balance. Microbial tryptophan metabolites can activate the aryl hydrocarbon receptor (AHR), which inhibits inflammatory responses and improves the skin epidermal barrier. The gut microbiota also utilizes the nutrients L-carnitine and choline, both dietary trimethylamines (TMA, red box), as carbon and energy sources. These dietary sources can be converted by the microbiota to TMA, which is then oxidized to TMAO by flavin monooxygenases (FMO). TMAO is a molecule involved in the development of cholesterol and cardiovascular disease [86]. The blue arrow pointing down means a decrease. Created with BioRender.com.

**Figure 3 ijms-24-14322-f003:**
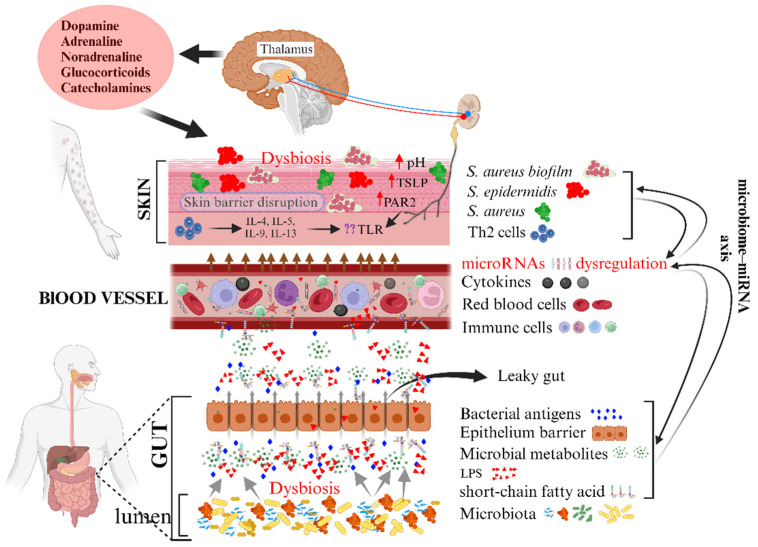
This figure shows the complicated relationship between the gut microbiota, microRNAs (miRNAs), and atopic dermatitis (AD). Different types of miRNAs and metabolites produced by the gut microbiota influence each other, affect the regulation of epithelial dysfunction, and contribute to the development of AD. Host miRNAs play a critical role in maintaining microbial balance in the gut and form the ‘microbiome–miRNA axis’. In addition, dysbiosis of the gut microbiota can lead to a leaky gut, allowing microbiota-derived metabolites and proinflammatory cytokines to circulate and reach the skin, leading to disruption of the skin barrier and activation of itch-causing factors of AD. This schematic model also suggests potential interactions between miRNAs, the gut microbiota, gut barrier integrity, and the immune system and highlights the multifaceted nature of these relationships in AD. The red arrow pointing up means an increase. Created with BioRender.com.

## Data Availability

Not applicable.

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
