# Peer review of "The Interaction between the Host Genome, Epigenome, and the Gut–Skin Axis Microbiome in Atopic Dermatitis"

_ijms, 2023, doi:10.3390/ijms241814322_

Round 1

Reviewer 1 Report

This review by Pessoa, Clissa and Sanabani merits consideration for publication in IJMS. The section on host miRNAs and microbiome, especially its metabolites, is interesting and is important to mention in the field of autoimmunity, such as atopic dermitis (or eczema), as host genetics does not determine disease outcome, as the authors rightfully state. 

Points

1. Figure 1 is of low resolution. A higher resolution figure should be included. The legend is also inappropriate. There should be a clear message (description) and it should stand alone, and not a continuation of the main text. The legend should not refer to the figure itself.

2. Add an additional figure illustrating the potential association between microbiome metabolites and miRNA in the possible prognosis of atopic dermatitis (section 6.1). 

3. I found the sections to be coherent and they provide an effective overall overview of the subject.

4. I understand the information on this interaction between host atopic dermatitis is limited, Pubmed the field really only took off in 2016 (Pubmed), so the number of references in this review seems to be adequate.

Author Response

Queries/critiques are in black Times Roman font. Responses follow in blue italic Times Roman font. Revised prose from within the manuscript is in red Arial font.

Review#1

This review by Pessoa, Clissa and Sanabani merits consideration for publication in IJMS. The section on host miRNAs and microbiome, especially its metabolites, is interesting and is important to mention in the field of autoimmunity, such as atopic dermitis (or eczema), as host genetics does not determine disease outcome, as the authors rightfully state. 

Points

  1. Figure 1 is of low resolution. A higher resolution figure should be included. The legend is also inappropriate. There should be a clear message (description) and it should stand alone, and not a continuation of the main text. The legend should not refer to the figure itself.

Reply: Thank you for your time and effort in reviewing our manuscript. We have provided a new Figure 1 at a much higher resolution of 600 DPI to improve clarity and detail. At your request, we have also rewritten the legend to Figure 1. The new legend now contains a clear and concise description that stands independently of the main text. We have also made sure that the legend does not refer to the figure itself, as you suggested.

  1. Add an additional figure illustrating the potential association between microbiome metabolites and miRNA in the possible prognosis of atopic dermatitis (section 6.1).

 Reply: We appreciate your suggestion to add an additional figure illustrating the potential association between microbiome metabolites and miRNA in the prognosis of atopic dermatitis. We have prepared a new figure in response to your recommendation. 

  1. I found the sections to be coherent and they provide an effective overall overview of the subject.

Reply: Thank you for your feedback! We are glad to hear that you found the sections coherent and that the overview of the topic was effective. We very much appreciate your positive comments

  1. I understand the information on this interaction between host atopic dermatitis is limited, Pubmed the field really only took off in 2016 (Pubmed), so the number of references in this review seems to be adequate.

Reply: Thank you for your feedback! Your insights are greatly appreciated.

Reviewer 2 Report

The authors have written a review on how gut microbiome potentially plays a role in the development of atopic dermatitis. The review is comprehensive, easy to comprehend and might serve as a good starting point for someone interested in pursuing this field. I have few minor comments that might add to the value of the manuscript:

1. There are sections such as 6.1 and 6.2 in which authors have not mentioned in details which specific gut bacteria contribute to these effects. This omission  is also pretty glaring in context of SCFAs. Authors have repeatedly come back to SCFAs as a potential mechanism in the pathogenesis of atopic dermatitis via gut microbiome alteration, but have not specified examples of bacteria other than Coprococcus. Which bacteria alter SCFAs?

2. What metabolites are produced by probiotic bacteria that can contribute to the pathogenesis of AD?

3. Please italicize the names of bacterial taxons.

Author Response

Queries/critiques are in black Times Roman font. Responses follow in blue italic Times Roman font. Revised prose from within the manuscript is in red Arial font.

Review#2

The authors have written a review on how gut microbiome potentially plays a role in the development of atopic dermatitis. The review is comprehensive, easy to comprehend and might serve as a good starting point for someone interested in pursuing this field. I have few minor comments that might add to the value of the manuscript:

Reply: Thank you for your thoughtful review and positive feedback. We sincerely appreciate your comments and suggestions, and we have carefully considered them in the revision of our manuscript to enhance its value.

  1. There are sections such as 6.1 and 6.2 in which authors have not mentioned in details which specific gut bacteria contribute to these effects. This omission  is also pretty glaring in context of SCFAs. Authors have repeatedly come back to SCFAs as a potential mechanism in the pathogenesis of atopic dermatitis via gut microbiome alteration, but have not specified examples of bacteria other than Coprococcus. Which bacteria alter SCFAs?

Reply: Thank you for your valuable feedback. We agree with your suggestion and we have made revisions to our manuscript and have included this information on page 4 of the revised version. We hope that these additions enhance the clarity and depth of our discussion on the mechanisms underlying atopic dermatitis via gut microbiome alterations. The relevant information in the “Environmental factors, the gut microbiome, and AD” section now reads:

“Coprococcus are bacteria that produce SCFAs, along with Bifidobacterium, Blautia, Eubacterium, and Propionibacterium, all of which have been detected in lower amounts in individuals with AD compared to those without AD” page 4, line 156-158

"In the large intestine of healthy humans, acetate is produced by Bifidobacteria and certain Firmicutes and Bacteroidetes species [68], while propionate is produced by Bacteroidetes, Negativicutes (Firmicutes strain) and Lachnospiraceae [69]. Butyrate is produced by Faecalibacterium prausnitzii, Eubacterium rectale, Eubacterium hallii [70]" Page 4, line 163-167.

  1. What metabolites are produced by probiotic bacteria that can contribute to the pathogenesis of AD?

Reply: We have included detailed information on this subject on page 7 (258-299) of the revised version of our manuscript. We have also included a new figure (Figure 2) illustrating the three major microbial metabolites. We believe that these additions improve the comprehensibility and clarity of our manuscript and provide our readers with a more comprehensive understanding of the metabolites associated with probiotic bacteria and their potential role in pathogenesis AD.

Please italicize the names of bacterial taxons.

Reply: We appreciate your input. The names of the bacterial taxons have been italicized as per your recommendation.
